# Effectiveness of an Evidence-Based Practice Educational Intervention among School Nurses

**DOI:** 10.3390/ijerph17114063

**Published:** 2020-06-07

**Authors:** Pei-Lin Hsieh, Sue-Hsien Chen

**Affiliations:** 1School of Nursing, Chang Gung University of Science and Technology, Chang Gung Medical Foundation, Taoyuan City 33303, Taiwan; 2Department of Nursing Management, Chang Gung Medical Foundation, School of Nursing, Chang Gung University, Taoyuan City 33303, Taiwan; a66998@cgmh.org.tw; 3Chang Gung University Science and Technology, Taoyuan City 33303, Taiwan

**Keywords:** evidence-based practice, mobile learning technology, flipped classroom, school nurse

## Abstract

The evidence-based practice (EBP) approach to high-quality care is designed to improve patient outcomes. However, little research has been conducted to determine how EBP is adopted and implemented among school nurses in Taiwan. This study evaluated an EBP training program implemented among school nurses in Taiwan to determine whether and how effectively it improved their knowledge levels, attitudes, skills, and self-efficacy. A pretest-posttest research design was employed. Participants were recruited via convenience sampling from among 193 primary schools throughout Tao-Yuan and New Taipei City in Taiwan. The EBP program implemented both mobile learning technology and the flipped classroom format. The learning outcomes were evaluated before, immediately after, and at a 3-month follow-up. In this regard, the data were collected using the School Nurse Evidence-Based Practice Questionnaire. The results showed that the participant scores for the items of knowledge and self-efficacy significantly increased over the study period. Somewhat differently, scores for the skill items significantly increased from the pre-test to the immediate post-test, but significantly decreased from the immediate post-test to the final follow-up. Overall, however, the EBP program led to marked improvements in knowledge, skills, and self-efficacy. These findings can help guide the development of a creative evidence-based school nursing curriculum.

## 1. Introduction

Evidence-based practice (EBP) has become an important trend in the health care field [1]. For instance, it can improve the quality of patient care, satisfy societal demands for certain types of care, and reduce care costs, thus making it a notably useful model for addressing 21st-century medical care needs [2]. As such, there have been strong efforts to implement EBP in Taiwan. In Taiwan, EBP has been included in foundation nursing education in Bachelor’s degrees as a core competence since 2013 [3], and it is now a core competency for clinical nursing staff [3,4,5,6]. EBP is a recent new trend in Taiwanese nursing education, and in Hsieh’s national survey of 1200 primary school nurses, 81% of them responded that they had not received an EBP-related course [7]. Nowadays, new EBP is a new era in clinical care, and therefore we expect to see the implementation of EBP and promotion campaigns for school nurses. Further, the Ministry of Education Taiwan [8] has developed an evidence-based second-generation health promotion school mechanism that prioritizes the utilization of evidence from successfully established school health-promotion programs as the basis for executing other such programs and evaluating school health policies. As executors in this process, school nurses play a vital role during implementation. A national survey was conducted among 247 school nurses in the United States to determine the status of EBP implementation in the school nursing environment, thus identifying that the participants had insufficient EBP skills, lacked necessary support systems, and had insufficient resources [9]. The participants also mentioned the paucities of EBP skills, resources, structures, and support mechanisms for schools when developing EBP. Since children and adolescents are valuable human assets for any country, it is important for schools to support them by adopting flexible strategies to promote children’s health. Studies have already shown that school-based health promotion programs play major roles in establishing healthy behavior [10,11]. Because school nurses are key persons in this regard, they are expected to have appropriate knowledge and apply established research findings when managing children’s needs during health education and when implementing prevention programs [12]. It is therefore vital for school nurses to recognize the importance of and apply EBP in their schools.

The EBP program was comprised of the five following steps: (1) ask a question, (2) find information/evidence to answer the question, (3) critically appraise the information/evidence, (4) integrate the appraised evidence using your own clinical expertise and patient preferences, and (5) evaluate [13]. Many medical centers in Taiwan have established evidence-based practice centers, organized national EBP seminars and workshops, created webpages, and formed promotion groups to facilitate EBP education and training [14,15,16]. Lin [17] even incorporated EBP into hospital nursing guidelines and a nursing auditing system. Further, Taiwanese scholars have advocated for the inclusion of EBP into basic education programs for new nursing recruits. Many have also recommended including EBP performance ability as an evaluation criterion for nursing staff, formulating different levels of EBP reports, and establishing a corresponding education and training mechanism [18,19,20]. However, few studies have investigated EBP performance levels among school nurses. As such, this study was conducted to investigate the issue and draw attention to the related research gap.

Education and training can improve EBP skills and knowledge among nursing staff, thereby facilitating clinical implementation. The current literature shows that EBP education and training approaches now include evidence-based training courses [21,22,23], lectures [24,25], seminars [23,26], journal clubs [27,28], practical exercises [16,29], and group counseling [23,28]. Studies have already provided evidence that sufficient EBP education and training programs should include both classroom teaching and practical exercise, as the latter component demonstrates better effectiveness in improving student knowledge levels, skills, and attitudes [1,30]. Different approaches can also be adopted based on the specific learning requirements of individual students (e.g., self-learning and question-oriented learning), which further improves learning effectiveness [21]. The literature suggests that core competencies can substantially improve in the area of evidence-based knowledge by combining blended learning and flipped-classroom methods via the implementation of multiple classroom activities, including case analyses, scenario-based learning, and practical exercises [31,32]. Furthermore, several related studies have demonstrated that blended learning significantly improved skill development and self-reported learning abilities among students [33,34,35,36]. To a certain extent, the flipped classroom is another format of blended learning, in which teachers interact with students rather than simply lecturing. Research has shown that the flipped classroom environment is positively related to learning effectiveness, especially in the areas of self-learning and problem-solving skills [37,38,39].

Overall, while EBP has fully been implemented among nurses working in the clinical practice setting, such application is wholly insufficient in the context of school nursing. The EBP training program investigated in this study was based on Hsieh, Chen, and Chan’s findings about related requirements for school nurses [7]. In this regard, the program differed from clinical EBP programs in one major area. That is, all the basic theories and case analyses were based on school nursing topics, thereby tailoring the program to facilitate learning motivation among school nurses. This study also attempted to incorporate mobile technologies and teaching resources that were specifically designed to increase EBP abilities among this demographic. This study aimed to evaluate the effectiveness of the EBP training program on school nurses’ knowledge, attitudes, skills, and self-efficacy.

## 2. Materials and Methods

### 2.1. Study Design

This study implemented a pretest-posttest research design. All the methods were approved by the Ethical Review Committee of Chang Gung Hospital, Linkou, Taiwan (approval number: 103-3678C). Further, all the participants provided their written informed consent prior to participation.

### 2.2. Participants

A total of 401 elementary school nurses were recruited from Tao-Yuan and New Taipei City in Taiwan via convenience sampling. Three hundred and thirty-six school nurses enrolled in the EBP training program, and 193 of them completed the 18 h training program. The inclusion criteria were set as follows: school nurses working in elementary schools and completing 18 h of the EBP training program. All others were excluded.

### 2.3. Intervention

This study received approval from both the Ministry of Education Taiwan and the Association of Chinese School Health Nursing. Further, the Association of Chinese School Health Nursing provided a list of school nurses working in both Tao-Yuan City and New Taipei City. Letters seeking permission and consent forms were sent and subsequently signed by both the Ministry of Education and the Association of Chinese School Health Nursing. The Ministry of Education then released a formal government document to inform elementary school nurses in Tao-Yuan City and New Taipei City that this study had received official support.

The intervention consisted of an evidence-based school nursing program that mainly focused on the five steps listed in the introduction (i.e., ask a question, find information/evidence to answer the question, critically appraise the information/evidence, integrate the appraised evidence via one’s own clinical expertise and patient preferences, and evaluate). The training program was a combination of several teaching strategies lectures, group activities that required the participants to be interactive, and supervision. The lectures were based on mobile learning technology, the flipped-classroom format, and the traditional in-class format. In this regard, approximately eight hours of class time were conducted online, while 10 h were conducted via the traditional in-class (face-to-face) format, and two hours consisted of online supervision support. We focused on outcome measures for evaluating the EBP training program, including knowledge, attitudes, skills, and self-efficacy at three points (i.e., baseline, post-training, and at 3-month follow-up). Table 1 provides a general outline of the educational components.

#### The Flipped Classroom Technology Implementation 

The flipped classroom was divided into two learning environments: outside and inside the classroom. Outside the classroom was based on mobile learning and inside the classroom was based on group activities. Online lessons were recorded, and each lesson consisted of 6 units (Table 1). Each unit provides a 20 min video. Five quizzes were provided in the end of each online lesson to check the participants’ comprehensive reading. A set of six video-lessons could be accessed for two weeks. All the videos were accessible in the Share Course platform to support self-directed learning such as online status, the number of videos viewed, and the total number of videos viewed. Reminders were sent to participants telling them that they were running out of time and encouraging them to complete the lessons. The video formats were supported by various browsers, for example, Google, Android, and iPhone. The participants could easily access the learning materials on devices.

Following the online lesson, a lesson inside the classroom was offered and participants were divided into 24 groups; each group had 7–8 members. An instructor was in charge of 4 groups, and a total of 6 instructors were involved in the process. The instructors were qualified by the Taiwan Evidence-Based Medicine Association. The instructors took particularly controversial quiz questions from the online material, and participants reflected on them and then discussed them further. Afterwards, participants reviewed a case study concerning a school health-based scenario and applied what they learned in the online lessons; the group discussed how they would tackle the problem and what solution they would prepare. Each group debriefed and presented their solution. The instructor elicited responses from all the members of the groups and began to engage participants in a wider discussion, demonstrating the many different perspectives. This process took approximately three hours.

### 2.4. Instruments

This study used the previously established School Nurse Evidence-Based Practice (SNEBP) Questionnaire [7], which is comprised of five sections (demographic data, knowledge levels, attitudes, skills, and self-efficacy). The demographic data included age, education level, the duration of employment as a school nurse, participation in research-based and EBP training programs, and the frequency of reading journal articles. The knowledge section consisted of five multiple-choice items (true/false/I do not know); an incorrect or “I do not know” answer was assigned a score of 0, while a correct answer was assigned a 1. Here, higher scores indicated higher levels of EBP knowledge. The content validity index (CVI) value was 0.83 and the value of internal consistency (Cronbach’s alpha) was 0.89. The attitudes section consisted of 12 items. All the items were answered on a 5-point Likert-scale, ranging from 1 (completely disagree) to 5 (completely agree). Higher scores indicated better attitudes. The content validity index (CVI) value was 0.97, and the value of internal consistency (Cronbach’s alpha) was 0.82. The self-efficacy section consisted of 12 items. All the items were answered on a 5-point Likert scale, ranging from 1 (no confidence) to 5 (complete confidence). Higher scores indicated a better self-efficacy. The content validity index (CVI) value was 0.87, and the value of internal consistency (Cronbach’s alpha) was 0.84. The skill section consisted of 10 items. All the items were answered on a 5-point Likert scale, ranging from 1 (completely disagree) to 5 (completely agree). Higher scores indicated better skills. The content validity index (CVI) value was 0.97, and the value of internal consistency (Cronbach’s alpha) was 0.94.

All the negatively worded items were reverse-coded. The construction of the SNEBP is detailed in a previous study that also established reliability, validity, and construct validity [7].

### 2.5. Statistical Analyses

All the statistical analyses were conducted using IBM SPSS Statistics for Windows, Version 26 (SPSS Inc., Chicago, IL, USA). Descriptive statistics (percentages of frequencies, means, and standard deviations) were calculated to describe the distributions of responses for each item. Knowledge, attitudes, skills, and self-efficacy were assessed using a questionnaire that was administered at three time points (baseline, post-training, and at 3-month follow-up), while the program effectiveness was analyzed using a repeated-measures ANOVA. *P* values of less than 0.05 were considered statistically significant for all tests.

## 3. Results

### 3.1. Participant Characteristics

A total of 193 elementary school nurses from either Tao-Yuan or New Taipei City in Taiwan participated in this study. All the participants were female. The mean participant age was 35.7 years (range of 26–58), while the mean duration of employment as a school nurse was 10.1 years (range of 1–20). Table 2 provides a basic outline of the participant demographic data. As shown, most had Bachelor’s degrees (72%), while more than 80% had not pursued advanced studies (80.3%). Furthermore, most had no experience with research programs (62.7%) and had not attended EBP training courses (64.8%). Many did not frequently read journal articles (68.9% had only done so several months prior, while 5.2% had not done so at all during the previous year). The main motivations for attending EBP training were personal interest (47%) and job requirements (35.2%). Table 3 shows the top five training courses attended during the previous year.

### 3.2. The Learning Effectiveness of EBP Training Over Time

Participants took the SNEBP questionnaire at all three time periods (baseline, post-training, and at 3-month follow-up) for the purpose of conducting comparisons. A repeated-measures ANOVA showed that the mean scores for both knowledge (F (2, 211.07) = 65.36, *p* < 0.001) and self-efficacy (F (2, 160.57) = 33.19, *p* < 0.001) significantly increased from baseline to follow-up. Next, a Bonferroni post hoc test revealed that knowledge scores increased by an average of 0.75 at post-training (*p* < 0.001), then further increased by 0.37 at follow-up (*p* < 0.001). Similarly, the self-efficacy scores increased by an average of 0.3 at post-training (*p* < 0.001), then further increased by 0.13 at follow-up (*p* < 0.001).

However, the skill factor showed different results. The mean scores for skill (F (2, 207.67) = 58.24, *p* < 0.001) varied significantly between baseline, post-training, and follow-up. Interestingly, the Bonferroni post hoc test revealed that the skill score significantly increased from baseline (mean = 3.73; SD = 0.58) to post-training (mean = 3.80; SD = 0.62), but then significantly decreased (by an average of 0.1) between post-training and follow-up (*p* < 0.001).

Finally, the scores for attitudes about EBP were not statistically significant at any time period. Table 4 shows descriptive statistics for the learning effectiveness of the EBP training program, while Table 5 provides a summary of the repeated-measures ANOVAs for each scale at the three time points.

## 4. Discussion

Unlike traditional classroom education methods that prioritize lecturing, the flipped classroom environment implements a learner-centered design. Prior to this, students gained self-learned knowledge through basic memory techniques and understanding. However, teachers who focus on guidance through innovative applications, analyses, evaluations, and the creation of higher-level experiences have found that students can learn to think at higher levels, which increases learning effectiveness [40,41,42]. In this study, the participants significantly improved their knowledge levels, skills, and self-efficacy based on follow-up evaluations after completing an EPB program. This indicates that implementing the flipped classroom environment during EBP training can successfully help school nurses achieve superior learning outcomes. Indeed, these findings are consistent with those of previous studies reporting that the flipped classroom was effective during training programs aimed at nurses and medical students. For instance, students can utilize their time during synchronous lessons to exchange views and solve questions via discussions held with teachers and other students while applying new practical knowledge through activities assigned by instructors, thereby improving the overall program effectiveness [36,43].

This study also found that the EBP skills were significantly higher immediately after the EBP training program (*p* < 0.05), but significantly decreased (from 3.80 to 3.70) based on the final follow-up results. After intervention, the school nurses might have felt that they were alone, without any requirements to apply EBP and the support that they had revived during the intervention phase. Possibly, they were also faced with barriers that hindered them applying EBP. In this study, we did not aim to identify EBP barriers. However, lack of time, lack of support system, lack of relevant resources, and lack of interest are frequently reported as barriers among nurses [44,45]. These barriers could explain why EBP skills were not sustained at follow-up. A previous study produced similar results, reporting that the most significant challenge in EBP implementation was the large gap between the application of knowledge from research sources and actual clinical practice [46]. It is essential to find ways of ensuing long-term change in EBP skills, and further studies need to take these barriers into account

In terms of self-efficacy when implementing EBP, scores significantly increased across all three time periods. Previous studies indicated that the self-efficacy of EBP was significantly correlated with education level and the duration of employment [7,45]. In our study, many school nurses had advanced degrees and the mean duration of employment as a school nurse was 10.1 years; these participant demographics may explain why the self-efficacy increased over time. Further, flipped classroom teaching gives students more opportunities to think critically [37]. The literature also shows that self-confidence can increase via this method [34,47]. However, 64.8% of this study’s participants had not attended any EBP training courses during the previous year, during which time very few relevant EBP courses had been included as part of their in-service training programs. Further, when asked about their motivation to participate in EBP training courses, 47.7% indicated personal interest, while 35.2% indicated that it was demanded by their job. These results show that it is urgent to conduct EBP training programs for school nurses. Although they are the only medical professionals working in schools, these school nurses lack access to the EBP training resources that nurses who work at medical institutions use. Relevant authorities (e.g., school nurse associations) should therefore address this issue by designing a series of training programs that are suitable for school nurses, thereby facilitating the implementation of EBP in the school nursing environment.

### Limitations

This study had some limitations. All the participants were recruited from either Tao-Yuan or New Taipei City in Taiwan, which diminishes its generalizability. First, the self-report questionnaire may not provide accurate information, so outcome measures such as an individual written assignment should be used to ensure the integrated learning of EBP. Secondly, school nurses must overcome their reliance on traditional classroom teaching and be willing to accept the responsibility for self-learning that comes with a flipped class. However, the inclusion criteria in this study was participants who completed 18 h of an EBP training program. Finally, there were no male school nurses involved, and a gender bias may exist. In Taiwan, the total number of elementary school nurses are 2,679, and there are 97 male school nurses (3.62%) [48]. The *number of male school nurses* is relatively *small, and* it was not easy to involve them in the study.

## 5. Conclusions

Post-training results showed that mobile learning technology and flipped-classroom support led to marked improvements in EBP knowledge, skills, and self-efficacy. However, the mean score for skills dropped based on the final follow-up results. Outcome measures, such as an individual written assignment or a case report, should be used to improve the long-term effects of EBP competencies through proper training. Further, health educators should play supporting roles during EBP implementation and help school nurses establish relevant routines.

## Figures and Tables

**Table 1 ijerph-17-04063-t001:** Outline of the evidence-based practice (EBP) training program.

Sections	Classroom Learning	Online Learning	Hours
The concept of EBP Instruction of EBP The five EBP steps	V		2
Ask: PICO question PICO framework Question statement	V		2
Development of a search strategy Search strategy Eligibility criteria Screening and data extraction		V	2
Databases for EBP research Electronic literature search strategies Practice	V		3
Supervision		V	1
Level of evidence A variety of evidence types and levels The characteristics of evidence		V	2
Appraisal of evidence Critical appraisal tools Reporting guidelines for EBP		V	2
Integration of evidence Apply all five steps to a school health scenario		V	2
Supervision		V	1
Oral presentation Apply the evidence in school health practice and present the report	V		3

EBP: Evidence-based practice; PICO: patient, intervention, comparator and outcome; V: the way of learning.

**Table 2 ijerph-17-04063-t002:** Participant demographics (*N* = 193).

Variables	*N*	%
Highest degree		
Master’s degree	38	19.7
Bachelor’s degree	139	72.0
Diploma	16	8.3
Continuing education		
No	155	80.3
Yes	38	19.7
Participation in research programs		
No	121	62.7
Yes	72	37.3
Participation in EBP training courses		
No	125	64.8
Yes	68	35.2
Frequency of reading journal articles in the previous year		
Never	10	5.2
Several months	133	68.9
At least every month	26	13.5
Every two weeks	14	7.3
At least every week	10	5.2
Motivation for attending EBP training (multiple choice)		
Personal interests	92	47.7
Job requirements	68	35.2
Nursing; continuing education requirements	37	19.2
Others	12	6.2

EBP: Evidence-based practice.

**Table 3 ijerph-17-04063-t003:** The top five training courses from the previous year (multiple-choice) (*N* = 193).

Courses	*N*	*%*	Ranking
Emergency care	82	42.5	1
Health promotion program	72	37.3	2
Infectious disease prevention and management	68	35.2	3
Health information management	50	25.9	4
Crisis management	38	19.7	5

**Table 4 ijerph-17-04063-t004:** Descriptive statistics for the learning effectiveness of the EBP training program over time (*N* = 193).

Item	Mean	SD	*p*
Knowledge of EBP			
Baseline	2.71	1.10	0.000 ***
Post-training	3.46	0.86	
Follow-up	3.83	0.83	
Attitudes about EBP			
Baseline	3.73	0.58	0.09
Post-training	3.85	0.60	
Follow-up	3.86	0.61	
Skills in EBP			
Baseline	3.26	0.77	0.000 ***
Post-training	3.80	0.62	
Follow-up	3.70	0.50	
Self-efficacy in implementing EBP			
Baseline	3.42	0.76	0.000 ***
Post-training	3.72	0.79	
Follow-up	3.85	0.75	

EBP: Evidence-based practice; SD: standard deviation; *** *p* < 0.001.

**Table 5 ijerph-17-04063-t005:** Summary of the repeated-measures ANOVAs for each scale.

Source of Variation	SS	df	MS	F	Multivariate Eta Squared
Knowledge of EBP	171.33	2	42.60	65.36 ***	0.96
Attitudes about EBP	96.34	2	0.64	4.10	0.98
Skills in EBP	111.94	2	10.41	58.24 ***	0.99
Self-efficacy in implementing EBP	177.14	2	6.43	33.19 ***	0.21

EBP: evidence-based practice; SS: sum of squares; MS: mean square; df: degrees of freedom; *** *p* < 0.001.

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
