# Peer review of "Effectiveness of an Evidence-Based Practice Educational Intervention among School Nurses"

_ijerph, 2020, doi:10.3390/ijerph17114063_

Round 1
Reviewer 1 Report
This paper discusses the effectiveness of an evidence-based practice educational intervention among school nurses.
The manuscript is well organized, however, it needs a revision in the writing. The study needs to highlight its main contribution concerning novelty.
Line 108, the authors say "The program was based on mobile learning technology and the flipped-classroom format." What were the limitations of this methodology? Have all students been reached? Which percentage of nurses could not attend the study and why?
The authors should better describe the data used in this research.
Author Response
Response to Reviewer 1 Comments
Reviewer 1
Point 1: Line 108, the authors say "The program was based on mobile learning technology and the flipped-classroom format." What were the limitations of this methodology?
Response 1: Thank you for your comment. The limitation of flipped-classroom format was added in line 270-276.
Point 2 Have all students been reached? Which percentage of nurses could not attend the study and why? The authors should better describe the data used in this research.
Response 2: Thank you for your comments. A total of 401 elementary school nurses were recruited from Tao-Yuan and New Taipei City in Taiwan via convenience sampling. Three hundred and thirty-six school nurses enrolled in the EBP training program, and 193 of them completed the full 18-hour training program. The inclusion criteria were set as follows: school nurses working in elementary schools and who completed 18 hours of EBP training program. We have revised and made it clearer in line 102-106.

Reviewer 2 Report
In this study, Hsieh and Chen continue their work on school-based nurses employing evidence-based practice (EBP) in their school environments. The original study surveyed nurses to discover a deficit in EBP perceptions and skills, and this continuation seeks to reduce that deficit through a particular intervention. As school-based health initiatives have the potential to affect the entire population, this is an important outcome to measure.
Strengths of the study include a high rate of participation from the target population and use of a previously-validated tool to measure the outcome. Weaknesses include a lack of detail in describing the intervention, a somewhat vaguely stated primary outcome, and a discussion that spends much time answering a question that was not asked earlier in the manuscript.
I will admit that I had some difficulty in reviewing this article. Starting at the title, I had questions I hoped would be answered in the manuscript that were not. For example, "effectiveness" might be better linked explicitly to improvement in the 5 domains of the SNEBP questionnaire, similar to the title for the original study.
The intervention is not described adequately. The amount of time spent on each topic is listed, and the flipped classroom is described, but a flipped classroom usually implies learning activities, and these could be better described, especially if an interested reader wants to duplicate this intervention. Knowing that you spent 2 hours on answering a PICO question does not help me know where to begin and what to focus on. For some health professions, the topics on this list make an entire semester-long course that you have taught in 18 hours mixed online and in-person; I need to know more of what you did if I will be able to reproduce it. In your background, you list many activities that have been used in the past to teach nurses, including courses, journal clubs, seminars, and group activities. Did you also use these methods? If so, when and how often. Time for an entire group to participate in journal club may easily have used entire hours of your in-person time, so this is important.
This contributes to a more important point that nurses are typically not taught evaluation and application of medical literature in their nursing programs. Many of your nurses have advanced degrees, so I would expect them to be more familiar, but I would not expect nurses with only a bachelor's degree to have much exposure past on-the-job training. The uniqueness of this situation is not mentioned anywhere in the article. If nurses in Taiwan are typically taught this in schools, their previous education should be described. If they are not, describing your intervention in more detail is even more important.
The discussion is somewhat confusing in that it begins with the flipped classroom method. This is almost half of the entire discussion section by line count, yet the methods never identified the flipped classroom as an important outcome, and no measurements of the flipped classroom were reported in results. The discussion asserts that the flipped classroom helped, but that is an invalid statement in this study. Without a comparison, there is no way to know that the flipped classroom produced superior results to what may have been achieved through another method.
The discussion and conclusion continue to describe the improvement 4 of the 5 domains, but absent is a discussion that the decline in skills ("I can critically appraise articles," "I can summarize the results of multiple articles to guide healthcare decisions that are most beneficial to student") is directly contrary to the increase in self-efficacy ("I can critically appraise articles... to guide healthcare decisions," "I can integrate evidence and apply it in school nursing") over the same time period. The discussion should address this paradox.
I agree with the authors' self-identified limitation that they limited the outcome measure to completion of the survey, which is a low level of evidence especially for the more important outcome of actual skills. I do wish the authors had chosen a graded PICO or other assignment for their outcome. This or another suggestion should be repeated in the conclusions where potential future research is discussed. Also, the researchers say they "were limited," suggesting maybe using only the survey was not their choice. What were the reasons for using a survey instead of a more practical outcome?
This manuscript is mostly well-written, except that I think it is incomplete in some important areas. Also, the topic is important and interesting from the standpoint of many different stakeholders in the health care system. This overall line of research should be continued, I hope with better outcome measures and more success with long-term retention of skills and attitudes.
Author Response
Response to Reviewer 2 Comments
Reviewer 2
Point 1: In this study, Hsieh and Chen continue their work on school-based nurses employing evidence-based practice (EBP) in their school environments. The original study surveyed nurses to discover a deficit in EBP perceptions and skills, and this continuation seeks to reduce that deficit through a particular intervention. As school-based health initiatives have the potential to affect the entire population, this is an important outcome to measure.
Response 1: Thank you so much for giving us an opportunity to revise this manuscript. We did learn a lot form the three reviewers. Also, we tried very hard to revise the following comments. Thanks again for the valuable comments.
Point 2 Strengths of the study include a high rate of participation from the target population and use of a previously-validated tool to measure the outcome. Weaknesses include a lack of detail in describing the intervention, a somewhat vaguely stated primary outcome, and a discussion that spends much time answering a question that was not asked earlier in the manuscript.
Response 2: Thank you for your comments. The more detail intervention descriptions was added in line 118-121, 123-125, and 131-154. The discussion was revised and made it clearer in red colour.
Point 3: I will admit that I had some difficulty in reviewing this article. Starting at the title, I had questions I hoped would be answered in the manuscript that were not. For example, "effectiveness" might be better linked explicitly to improvement in the 5 domains of the SNEBP questionnaire, similar to the title for the original study.
Response 3: Thank you for your comments. We have revised the each comment and hopefully answered the reviewer’s questions. The revision is in red colour.
Point 4: The intervention is not described adequately. The amount of time spent on each topic is listed, and the flipped classroom is described, but a flipped classroom usually implies learning activities, and these could be better described, especially if an interested reader wants to duplicate this intervention. Knowing that you spent 2 hours on answering a PICO question does not help me know where to begin and what to focus on. For some health professions, the topics on this list make an entire semester-long course that you have taught in 18 hours mixed online and in-person; I need to know more of what you did if I will be able to reproduce it. In your background, you list many activities that have been used in the past to teach nurses, including courses, journal clubs, seminars, and group activities. Did you also use these methods? If so, when and how often. Time for an entire group to participate in journal club may easily have used entire hours of your in-person time, so this is important.
Response 4: Thank you for your comments. A flipped classroom was divided into two learning environments: outside and inside the classroom in this study. Outside the classroom was bas on mobile learning and inside the classroom was based on group activities. The flipped classroom intervention was added in line 131-154.
EBP has been included in bachelor’s degrees in foundation nursing education as a core competence since 2013 in Taiwan [1]. EBP is a new trend in Taiwanese nursing education and in Hsieh’s national survey of 1,200 primary school nurses, 81% of them responded that they have not received EBP related training courses [2]. Nowadays, EBP is a new era in clinical care and therefore we expect to see the implementation of EBP and EBP promotional campaigns for school nurses. Therefore, the basic of EBP was designed in this study for example, the characteristics of evidence used in EBP, basic terms and processes of EBP, these provide a basis for school nurses to adopt evidence-based practice in schools . We have added objectives in Table 1 to make it clearer.
Lectures, group activities and supervision were used in the training program and we have added these strategies in line 118-121.
Point 5: This contributes to a more important point that nurses are typically not taught evaluation and application of medical literature in their nursing programs. Many of your nurses have advanced degrees, so I would expect them to be more familiar, but I would not expect nurses with only a bachelor's degree to have much exposure past on-the-job training. The uniqueness of this situation is not mentioned anywhere in the article. If nurses in Taiwan are typically taught this in schools, their previous education should be described. If they are not, describing your intervention in more detail is even more important.
Response 5: Thank you for your comments. Even though many of school nurses have advanced degree, EBP is a new trend in Taiwanese nursing education. EBP has been included in bachelor’s degrees in foundation nursing education as a core competence since 2013 in Taiwan [1]. Based on the results of Hsieh’s study [2], the EBP program is tailored for school nurses in this study to increase EBP ability. We have added more explanation in introduction (line 34-40) and intervention (line 131-154) section to make it clearer.
Point 6: The discussion is somewhat confusing in that it begins with the flipped classroom method. This is almost half of the entire discussion section by line count, yet the methods never identified the flipped classroom as an important outcome, and no measurements of the flipped classroom were reported in results. The discussion asserts that the flipped classroom helped, but that is an invalid statement in this study. Without a comparison, there is no way to know that the flipped classroom produced superior results to what may have been achieved through another method.
Response 6: Thank you for your comments. In this study, the training program was combinations of several teaching strategies lectures, group activities that required participants to be interactive and supervision. Lectures were based on mobile learning technology, the flipped-classroom format and traditional in-class format. We focus on outcome measures for evaluating EBP training program including knowledge, attitudes, skills, and self-efficacy of the SNEBP. As the flipped classroom was a part of strategy, the measurements were included in the knowledge, attitudes, skills, and self-efficacy section of the SNEBP. We have added more explanation and revised it in the intervention, the flipped classroom technology implementation in red colour and the discussion section (line 231-232).
Point 7: The discussion and conclusion continue to describe the improvement 4 of the 5 domains, but absent is a discussion that the decline in skills ("I can critically appraise articles," "I can summarize the results of multiple articles to guide healthcare decisions that are most beneficial to student") is directly contrary to the increase in self-efficacy ("I can critically appraise articles... to guide healthcare decisions," "I can integrate evidence and apply it in school nursing") over the same time period. The discussion should address this paradox.
Response 7: Thank you for your comments. We have added discussion in line 240-245, 248-249, and 251-254.
Point 8: I agree with the authors' self-identified limitation that they limited the outcome measure to completion of the survey, which is a low level of evidence especially for the more important outcome of actual skills. I do wish the authors had chosen a graded PICO or other assignment for their outcome. This or another suggestion should be repeated in the conclusions where potential future research is discussed. Also, the researchers say they "were limited," suggesting maybe using only the survey was not their choice. What were the reasons for using a survey instead of a more practical outcome?
Response 8: Thank you for your comments. “Outcome measures such as an individual written assignment should be used to ensure integrated learning of EBP.” was added in limitations section (line 270-272). “Outcome measures such as individual written assignments or a case report should be used to improve the long-term effects of EBP competencies through proper training.” was added in conclusions section (line 285-287).
Point 9: This manuscript is mostly well-written, except that I think it is incomplete in some important areas. Also, the topic is important and interesting from the standpoint of many different stakeholders in the health care system. This overall line of research should be continued, I hope with better outcome measures and more success with long-term retention of skills and attitudes.
Response 9: Thank you for your comments.
References
- Chiang, L.C. A new vision of nursing: the evolution and development of
evidence-based nursing. J Nurs 2014, 61(4), pp. 85-94.
- Hsieh, P.L.; Chen, S.H.; Chang, L.C. School nurses’ perceptions, knowledge, and related factors associated with evidence-based practice in Taiwan. Int J Environ Res Pub Health 2018, 15(9), p. 1845.

Reviewer 3 Report
Dear authors
After review your manuscript I have a question about the instruments used, SNEBP questionnaire.
Will it be possible to include in your paper values of McDonald's Omega values to check the reliability of SNEBP?. When you cited Hsieh PL et al's manuscript (2018), and after reviewing it, I can't find the omega values and authors express only alpha's Cronbach values to define the questionnaire reliability. My point is if it possible that authors' give an explanation about the 5 items measured in SNEBP correspond to the same dimension or psychological construct?
On the other hand, an explanation of the sample calculation. What is the total number of school nurses working in elementary schools in Tao-Yuan and New Taipei City in Taiwan? This data is necessary to understand the statistical validity of your results and conclusions.
Related to sample I have a question about sex. Could you include data about women and men in your sample?. It will be so interesting to know the number of women (school nurses) that participate in the authors' study.
These data help to evaluate the existence of possible bias due to sex (feminized profession?) in data and conclusions derived from its.
In relation with study limitations, I recommend authors include a reference to:
- The self-report measures used to evaluate items from SNEBP.
- When including the sex, as a sociodemographic variable, comment (if appear) the possible existence of bias due to sex.
Author Response
Response to Reviewer 3 Comments
Reviewer 3
After review your manuscript I have a question about the instruments used, SNEBP questionnaire.
Point 1: Will it be possible to include in your paper values of McDonald's Omega values to check the reliability of SNEBP?. When you cited Hsieh PL et al's manuscript (2018), and after reviewing it, I can't find the omega values and authors express only alpha's Cronbach values to define the questionnaire reliability. My point is if it possible that authors' give an explanation about the 5 items measured in SNEBP correspond to the same dimension or psychological construct?
Response 1: Thank you for your comments. The validity was assessed by the content validity index (CVI) value and reliability was assessed by the Cronbach’s alpha. We have added the CVI and Cronbach’s alpha values in the knowledge, attitudes, skills, and self-efficacy section and make it clearer in line 160-172 and 174.
Point 2: On the other hand, an explanation of the sample calculation. What is the total number of school nurses working in elementary schools in Tao-Yuan and New Taipei City in Taiwan? This data is necessary to understand the statistical validity of your results and conclusions.
Response 2 : Thank you for your comments. A total of 401 elementary school nurses were recruited from Tao-Yuan and New Taipei City in Taiwan via convenience sampling. Three hundred and thirty-six school nurses enrolled in the EBP training program, and 193 of them completed the full 18-hour training program. The inclusion criteria were set as follows: school nurses working in elementary schools and who completed 18 hours of EBP training program. We have revised and made it clearer in line 102-106.
Point 3: Related to sample I have a question about sex. Could you include data about women and men in your sample?. It will be so interesting to know the number of women (school nurses) that participate in the authors' study.
These data help to evaluate the existence of possible bias due to sex (feminized profession?) in data and conclusions derived from its.
Response 3: Thank you for your comments. We have added ‘All participants were female.’ in line 186.
Point 4: In relation with study limitations, I recommend authors include a reference to:
- The self-report measures used to evaluate items from SNEBP.
- When including the sex, as a sociodemographic variable, comment (if appear) the possible existence of bias due to sex.
Response 4: Thank you for your comments. Limitations of self-report measures and gender bias are added in line 268-270, and 276-280.

Round 2
Reviewer 3 Report
Dear authors
Thanks for your reply, the changes included in your manuscript improve its quality.
Kind regards